# Graphene Oxide-Modified Epoxy Asphalt Bond Coats with Enhanced Bonding Properties

**DOI:** 10.3390/ma15196846

**Published:** 2022-10-02

**Authors:** Junsheng Zhang, Rui Wang, Ruikang Zhao, Fan Jing, Chenxuan Li, Qingjun Wang, Hongfeng Xie

**Affiliations:** 1MOE Key Laboratory of High Performance Polymer Materials and Technology, School of Chemistry and Chemical Engineering, Nanjing University, Nanjing 210093, China; 2Department of Chemistry, Texas A&M University, College Station, TX 77843-3255, USA

**Keywords:** epoxy asphalt, bond coat, graphene oxide, bonding strength, mechanical properties

## Abstract

The bonding strength of the bond coat plays an important role in the composite action between the wearing surface and the deck plate of the orthotropic steel deck system. Poor bonding results in the delamination of the wearing surface from the deck plate. Graphene oxide (GO) possesses outstanding mechanical and thermal properties, as well as impressive multifunctional groups, which makes it an ideal reinforcement candidate for polymer matrices. In this study, graphene oxide was used to improve the bonding strength and toughness of the epoxy asphalt bond coat (EABC). The dispersion, hydrophobicity, viscosity–time behavior, phase-separated morphology, dynamic mechanical properties, pull-off strength, shear strength and mechanical performance of GO-modified EABCs were investigated using various techniques. The inclusion of GO improved the hydrophobicity of the unmodified EABC. The viscosity of the unmodified EABC was lowered with the addition of GO during curing. Moreover, the allowable construction time for the modified EABCs was extended with the GO loading. The incorporation of GO enhanced the stiffness of the unmodified EABC in the glassy and rubbery states. However, graphene oxide lowered the glass transition temperature of the asphalt of the unmodified EABC. Confocal microscopy observations revealed that GO was invisible in both the asphalt and epoxy phases of the EABC. The inclusion of GO improved the bonding strength, particularly at 60 °C, and mechanical properties of the unmodified EABC.

## 1. Introduction

Because of the outstanding advantages of orthotropic steel deck (OSD) systems, such as the low life-cycle costs, suitability for prefabrication and standardization, low maintenance and improved work zone safety, they have been widely used in long-span bridges across large rivers or straits in many countries [1]. Apart from the OSD plate, wearing surfaces, including bituminous and polymer surfaces, are essential for OSD bridges since they provide an anti-skidding and comfortable surface, as well as corrosion protection, to deck plates, level out irregularities of the deck plate and increase the fatigue life of deck plates [2,3]. Additionally, to facilitate the composite action between the deck plate and the wearing surface, a bond coat is frequently used before the wearing surface is paved on the plate. In addition to preventing the wearing surface and the deck plate from shoving and delamination, using such a waterproof bonding layer also protects against the corrosion of deck plates and reduces tensile cracking in wearing surfaces. Due to the action of the less stiff composite between the deck plate and the wearing surface, which resist bending independently, together with the high flexibility of the thin deck plate, both the toughness of the bond coat and the bonding strength between the wearing surface and the deck plate need to be as high as possible compared to the binder used in the wearing surface. Otherwise, the soft bonding layer tends to result in the delamination of the wearing surface from the deck plate, especially in extreme cases [4,5].

Like the binder used in the wearing surface, bond coats can be divided into two classes: thermoplastic and thermosetting [6,7]. Due to their higher stiffness, most of the bond coats used on OSD plates are of the thermosetting type, which includes epoxy resins, polyurethane-epoxy and epoxy asphalt bond coats (EABCs) [8,9]. The epoxy asphalt binder and EABC used in bituminous surfacing have been extensively used in many long-span bridges in China and the United States [10,11,12,13,14]. Epoxy asphalt is mainly composed of asphalt and epoxy resin, whose major components are epoxy oligomers and a curing agent [15,16,17,18,19]. In EABCs, except for the component of epoxy oligomers, curing agents, asphalt and other additives are premixed as another individual component. After the two components of the EABC (comprising a bituminous curing agent and epoxy oligomers) are mixed and sprayed on a deck plate pre-coated with a layer of primer, hot bituminous surfacings, such as epoxy asphalt mixtures, are paved on the semi-liquid EABC layer. Finally, the bonding of the bituminous surface to the deck plate can be achieved through the curing reaction of epoxy resin. The mechanical and bonding properties of EABCs depend on not only the ratio of epoxy resin to asphalt but also the penetration grade of asphalt [9,20]. To improve the toughness and bonding strength of EABCs, many efforts have been made [21,22,23]. Sun et al. [21,22] used nanoclays to reinforce EABCs. Both pristine attapulgite and montmorillonite significantly improved the bonding and mechanical properties of the pure EABC. Huang et al. [23] reported that rubber particles surface-treated with a silane coupling agent greatly improved the low-temperature pull-off and shear strengths of the EABC.

Graphene is a two-dimensional allotrope of carbon with a honeycomb lattice. Due to its outstanding thermal, mechanical, electrical and optical properties, graphene has become an ideal nanofiller for composite materials [24]. However, the mass production of graphene has been too difficult to achieve thus far [25]. Therefore, graphene derivatives, such as reduced graphene oxide, graphene oxide (GO) and graphene nanoplates, have attracted extensive attention for the application of graphene-based materials because of their low cost and large-scale production [26]. Graphene oxide is obtained from the harsh oxidation treatment of graphite and is thus composed of oxygen functional groups, including epoxide, hydroxyl and carboxylic acid groups, as opposed to graphene [27]. Although these groups result in defects in the structure of graphene to some extent, they make graphene oxide stand out for a wide variety of applications because of their easy modification with other functional groups and their excellent dispersibility in many solvents [28,29]. Consequently, the modification of asphalt and epoxy resin with graphene oxide has gained more and more attention due to its unique structure and outstanding thermal, mechanical and electrical properties [30,31,32].

Recently, graphene oxide has also been used to modify epoxy asphalts [33,34,35]. Differential scanning calorimetry (DSC) curing kinetics revealed that both pristine- and hyperbranched-polyester-modified graphene oxides accelerate the curing reaction and lower the curing activation energy of epoxy asphalt [33]. Zhao et al. [34] reported that the inclusion of graphene oxide enhances the mechanical performance of epoxy asphalt binder. Si et al. [35] revealed that the incorporation of graphene oxide increases the pull-off strength, elongation and toughness of EABCs, whereas the tensile strength decreases. However, the dispersion of graphene oxide, hydrophobicity and shear strength of graphene oxide-modified EABCs were not reported. Additionally, the failure mechanism of adhesion was not interpreted.

In this study, graphene oxide was used to improve the toughness and bonding strength of EABCs. To achieve this goal, graphene oxide was first mixed with curing agents and asphalt. Then, the mixture was mixed with epoxy oligomers to prepare graphene oxide-modified EABCs. The effects of graphene oxide on the viscosity–time behavior, hydrophobicity, phase-separated morphology, dynamic mechanical properties, bonding strength and mechanical performance of the unmodified EABC were studied. In addition, the failure modes in pull-off tests were investigated.

## 2. Materials and Methods

### 2.1. Materials

Base asphalt was obtained from China Offshore Bitumen (Taizhou) Co., Ltd. (Taizhou, China). Table 1 provides an overview of base asphalt. The acid-based hardener was self-prepared in the laboratory. Bisphenol A epoxy oligomer with an epoxide equivalent weight of 192 g/eq was bought from Nantong Xingchen Synthetic Material Co., Ltd. (Nantong, China). Graphene oxide powder was provided by Suzhou Tanfeng Graphene Technology Co., Ltd. (Suzhou, China). The physical properties of graphene oxide powder are presented in Table 2.

### 2.2. Preparation of GO-Modified EABCs

To ensure the good distribution of GO in EABCs, a masterbatch of the GO-modified bituminous curing agent, composing of graphene oxide powder, curing agent and base asphalt, was prepared by a high-shear emulsifier at 120 °C for 15 min at a rotational speed of 4000 min^−1^. Then, the epoxy oligomer was introduced into the masterbatch and mixed at 120 °C for 5 min at a rotational speed of 200 min^−1^. Finally, the mixture of the uncured GO-modified EABC was poured into a Teflon mold with a diameter of 100 mm and a height of 5 mm and cured at 120 °C for 4 h. The weight ratio of the epoxy oligomer to the bituminous curing agent was 1:4.5. The weight percentages of graphene oxide in the modified EABCs were 0, 0.2, 0.5 and 1.0 wt%. Figure 1 depicts a schematic illustration for the preparation and characterization of GO-modified EABCs.

### 2.3. Methods

#### 2.3.1. X-ray Diffraction (XRD) Analysis

XRD analysis was conducted on a Shimadzu XRD-6000 X-ray diffractometer (Kyoto, Japan) with a Cu Kα source (λ = 1.54 Å). Graphene oxide and samples of the unmodified and GO-modified EABCs bonded on a slid holder were scanned at 5°/min in the range of 5−60°.

#### 2.3.2. Morphology

The phase-separated microstructures of the unmodified and GO-modified EABCs were observed on a laser scanning confocal microscope (LSCM, Zeiss LSM 710, Jena, Germany) with Ar+ laser light (488 nm). The average diameters and polydispersity index (*PDI*) of the dispersed phase in the confocal microscopy image were calculated by using professional image analysis software [11,36,37]:(1)$$d_{n}=\frac{\Sigma n_{i}d_{i}}{\Sigma n_{i}},$$
(2)$$d_{w}=\frac{\Sigma n_{i}d_{i}{}^{2}}{\Sigma n_{i}d_{i}},$$
(3)$$PDI=\frac{d_{w}{}^{}}{d_{n}}$$
where *d_n_* and *d_w_* are the number- and weight-average diameters. *n_i_* is the number of particles with a diameter of *d_i_*.

#### 2.3.3. Rotational Viscosity

The viscosity–time behavior was tested on a Brookfield rotational viscometer (RV, NDJ-1C, Shanghai, China). The uncured sample was placed into a chamber with a No. 28 spindle at 120 °C. The viscosity was recorded every 5 min at a rotational speed of 50 min^−1^ until it reached 5 Pa·s, according to ASTM D4402.

#### 2.3.4. Hydrophobicity

Contact angles were measured at room temperature on a contact angle meter (CAM, CAM 200, KSV, Helsinki, Finland). The contact angle was recorded within 1 min after deionized water (5 μL) was dripped on the surface of the cured sample. The average contact angle was obtained from at least five replicates for each measurement.

#### 2.3.5. Viscoelastic Properties

Dynamic mechanical analysis (DMA) was performed on a 01 dB-Metravib DMA + 450 instrument (Limonest, France) in tension mode. The test was conducted at a heating rate of 3 °C/min and 1 Hz from −50 °C to 100 °C.

#### 2.3.6. Pull-Off Adhesion Test

Pull-off strength was determined on a portable automatic adhesion tester (PosiTest AT-A, DeFelsko, Ogdensburg, NY, USA). The measurement was carried out at room temperature and at a pull-off rate of 0.7 MPa/s. To determine the pull-off strength, the uncured EABC was daubed on the surface of a polished Q345D steel plate (150 × 150 × 20 mm^3^) with a coating weight of 600 g/m^2^. Five dollies with a diameter of 20 mm were put on the surface of the EABC layer. Afterward, the steel plate was cured at 120 °C for 4 h. After the steel plate was cooled to room temperature, the testing areas around the dollies were separated by a core drill. Finally, a uniaxial load was employed perpendicularly to the dolly until the dolly was separated from the steel plate. Figure 2 shows the adhesion tester and testing configuration.

#### 2.3.7. Single-Lap Shear Test (SLST)

The shear strength was determined using the single-lap shear test as per ASTM D1002. The uncured EABC was daubed on the end of two steel plates (100 × 25 × 2 mm^3^). Then, the parts of the two plates coated with the bond coat (12.5 mm) were overlapped and cured at 120 °C for 4 h. After being cooled to room temperature, the joined parts of the two stainless-steel plates were pulled out using an Instron 3366 universal testing machine (UTM, Norwood, MA, USA) in tension mode at a crosshead speed of 50 mm/min. Five replicates were measured for each sample, and the average value was calculated. Figure 3 presents a schematic illustration of the sample for the single-lap shear test.

#### 2.3.8. Tensile Test

Tensile properties were tested on a UTM (Instron 3366, Norwood, MA, USA) according to ASTM D638. The test was conducted at room temperature at a crosshead speed of 200 mm/min. Five replicates were tested for each sample.

## 3. Results and Discussion

### 3.1. X-ray Diffraction Analysis

Figure 4 illustrates the XRD patterns of pristine GO and the unmodified and GO-modified EABCs. As shown in Figure 4a, a strong and sharp diffraction peak appears at 2θ = 11.46° in the XRD diffractogram of pristine GO, which is attributed to the (0 0 1) lattice plane of GO. Based on the Bragg equation, the interlayer distance of this peak is 0.77 nm. This broad interlayer distance is attributed to the existence of oxygen functional groups on the basal plane and at the edges of GO. For the unmodified EABC, as shown in Figure 4b, a broad diffraction peak is observed at 2θ = 19.63° due to the amorphous nature of both epoxy and asphalt. With the inclusion of GO, similar to the unmodified EABC, only one broad diffraction peak is observed at nearly the same degree, indicating that GO nanolayers are exfoliated in the epoxy asphalt during epoxy curing and thus well dispersed in the epoxy asphalt. The dispersion of GO nanolayers in both epoxy and the asphalt matrix is discussed in Section 3.5.

### 3.2. Hydrophobicity

Figure 5 shows the contact angles of the unmodified and GO-modified EABCs. Depending on the contact angle, the hydrophilicity of a material can be divided into four groups: (1) superhydrophilic (contact angle between 0° and 10°), (2) hydrophilic (contact angle between 10° and 90°), (3) hydrophobic (contact angle over 90°) and superhydrophobic (contact angle over 150°). In this case, the unmodified and GO-modified EABCs are hydrophobic, since the contact angles range from 98.2 to 102.6°. Furthermore, the water contact angles of GO-modified EABCs are higher than those of the unmodified EABC, which are 99.3°, 102.6° and 102.3°, respectively. When adding GO, the hydrophobicity of the unmodified EABC is improved. It is known that GO is hydrophilic due to the existence of oxygen functional groups, such as hydroxyl groups. However, these groups react with some functional groups of epoxy and asphalt during curing and thus improve the ability of the EABC to repel contact with water molecules.

### 3.3. Viscosity–Time Behavior

Figure 6 illustrates the viscosity–time curves of the unmodified and GO-modified EABCs at 120 °C. The viscosity of the unmodified EABC slightly decreases within 5 min, since the heat generated from the exothermal reaction of epoxy resin results in a viscosity decrease. After 5 min, as the molecular weight of epoxy resin increases, the viscosity increase quickly surpasses the viscosity reduction caused by heat. In addition, the exothermal heat is not easily dissipated from the sample and thus adds to the continued reaction. In this case, the viscosity increases with the curing time as the molecular growth of epoxy resin continues over time. After 15 min, the viscosity increases sharply due to the appearance of the gel point. The existence of GO lowers the viscosity of the unmodified EABC during curing. Further, for the modified EABCs, the viscosity decreases with the GO loading, indicating that the allowable construction time is extended with the GO loading.

Due to its more complicated construction technology, the allowable construction time of the epoxy asphalt mixture is much longer than that of the EABC. In this case, the reaction rate and viscosity of epoxy asphalt binders should be as low as possible. Otherwise, the mixture cannot become well compacted when the viscosity of the binder is higher than 3 Pa·s [38,39]. Therefore, to achieve a longer allowable construction time, the reaction rate of the epoxy asphalt binder is much lower than that of the EABC. According to the specification of GB/T 30598 [40], the time to reach 1 Pa·s for a warm-mix epoxy asphalt binder and an EABC should be over 40 and 10 min, respectively. The time to reach 1 Pa·s for the unmodified EABC is 14.1 min. However, with the inclusion of 0.2 wt%, 0.5 wt% and 1.0 wt%, the time to reach 1 Pa·s of the unmodified EABC extends to 16.2, 17.4 and 20.4 min, respectively. Therefore, the presence of GO extends the allowable construction time of the unmodified EABC. Further, the allowable construction time is extended with the GO loading.

It is known that GO is composed of a small quantity of oxygen functional groups, including epoxide, hydroxyl and carboxylic acid groups, which can react with both epoxy oligomers and curing agents, such as amines and carboxylic acids, of the epoxy resin [27,33,41]. Thus, these reactions compete with the curing reaction of epoxy resin. The curing reaction of epoxy resin undergoes chemical-controlled and diffusion-controlled stages [42]. Due to the tremendous difference in the quantity of reaction groups, epoxy curing is the dominant reaction in the chemical-controlled stage. In the diffusion-controlled stage, the reaction becomes confined. In this case, the hindrance effect of GO nanolayers limits the diffusion of the reactive groups and thus lowers the viscosity of the EABC during curing.

### 3.4. Dynamic Mechanical Properties

#### 3.4.1. Storage Modulus (E’)

Figure 7 presents the storage modulus–temperature curves of the unmodified and GO-modified EABCs. The addition of GO increases the E’ of the unmodified EABC in both the glassy and rubbery states. In the glassy transition, the storage moduli of GO-modified EABCs are similar to that of the unmodified EABC. However, for the modified EABC with 1.0 wt% GO, E’ is higher than that of the unmodified one throughout the whole temperature interval. Given that the storage modulus indicates the stiffness of a material [43], it can be concluded that the incorporation of GO improves the stiffness of the unmodified EABC in the glassy and rubbery states. In particular, the temperature of the OSD plate can reach over 60 °C in summer. Therefore, the stiffer bond coat in the rubbery state improves the bonding between the wearing surface and the deck plate.

#### 3.4.2. Glass Transitions

The loss modulus (E”) represents the energy dissipation ability of a material. For a polymer, E” is related to the morphology, glass transition, molecular motions, relaxation and so on [43,44]. As illustrated in Figure 8, all E”–temperature curves show two peaks in the temperature range of −30–30 °C, indicating that all EABCs have two glass transitions: the glass transition of asphalt at the lower temperature and the glass transition of epoxy at the higher temperature. Moreover, the glass transition peak of the unmodified EABC becomes more pronounced with the inclusion of GO. The glass transition temperatures (T_g_s) of both asphalt and epoxy obtained from the E”peak are summarized in Table 3. The inclusion of GO slightly decreases the T_g_s of both asphalt and epoxy in the unmodified EABC.

The loss factor (the ratio of E” to E’) represents the ability of a material to lose energy through internal friction and molecular rearrangements, which is independent of geometrical effects [45]. Similar to the E’–temperature curves, all tan-δ–temperature curves of the unmodified and GO-modified EABCs exhibit two glass transitions, as shown in Figure 9. The inclusion of GO slightly lowers the T_g_ of asphalt in the unmodified EABC. The T_g_ of epoxy shows a similar trend, except for the 1.0 wt% GO loading.

### 3.5. Phase-Separated Microstructures

Figure 10 depicts the fluorescence confocal microscopy images of the unmodified and GO-modified EABCs. The unmodified EABC exhibits a typical phase-separated morphology of epoxy asphalt (Figure 10a), in which black spherical particles (dispersed asphalt phase) are surrounded by yellow epoxy resin (continuous phase) due to the excitation of crosslinked epoxy. With the inclusion of 0.2 wt% GO, the spherical shape of asphalt particles in the unmodified EABC does not change, but the particle size becomes smaller (Figure 10b). However, for the modified EABCs with 0.5 wt% and 1.0 wt% GO, the shape of asphalt particles is observed to be more irregular, as shown in Figure 10c,d. It is worth noting that GO is invisible in all fluorescence confocal microscopy photographs of GO-modified EABCs due to the deep color of both dispersed and continuous phases.

To observe the dispersion of GO in the EABCs, transmittance confocal microscopy photographs are shown in Figure 11. However, GO is still unseen in both the dispersed and continuous phases of EABCs. As confirmed by the XRD results, epoxy curing results in the exfoliation of GO nanolayers, resulting in the uniform distribution of thinner GO nanolayers with a lighter color in the epoxy. Therefore, graphene oxide nanolayers are hard to distinguish from both asphalt and crosslinked epoxy resin.

Image-Pro Plus image analysis software is a useful tool for determining the particle size and distribution of dispersed asphalt phases in epoxy asphalts [46,47,48]. As shown in Table 4, the incorporation of 0.2 wt% GO decreases the average diameter and polydispersity of the unmodified EABC. The lower polydispersity, the more uniform the dispersion of the dispersed particles. Therefore, the distribution of asphalt particles becomes uniform with the presence of 0.2 wt% GO. The inclusion of 0.5 wt% and 1.0 wt% GO decreases the number-average diameter (*d_n_*) and increases the weight-average diameter (*d_w_*) and polydispersity of the unmodified EABC. Therefore, the distribution of asphalt particles in these GO-modified EABCs is more random than that in the unmodified EABC, as shown in Figure 10c,d and Figure 11c,d.

### 3.6. Pull-Off Strength

Figure 12 depicts the pull-off strengths of the unmodified and GO-modified EABCs. The presence of GO improves the pull-off strength, at both room temperature and 60 °C, of the unmodified EABC. At the same GO loading, the pull-off strength at room temperature is higher than that at 60 °C. In addition, the pull-off strength of the modified EABCs increases with the GO loading. The pull-off strengths at room temperature and 60 °C of the modified EABC with 1.0 wt% GO are 22% and 32% higher than those of the unmodified EABC. The improvement effect of GO on the adhesion properties is more pronounced at a higher temperature.

To evaluate the failure mechanism of the unmodified and GO-modified EABCs, the failure surfaces of the steel plates after pull-off tests were investigated. As presented in Figure 13a, the failure surface of the steel plate bonded by the unmodified EABC has two failure modes after pull-off tests at room temperature: adhesive failure and cohesive failure. Adhesive failure occurs at the interface of the steel plate and the bond coat, which is an indicator of the adhesion strength of the bond coat, whereas cohesive failure takes place within the coating, which is an indicator of the cohesive strength of the bond coat. Generally, cohesive failure is preferable since it indicates that the bonding between the steel plate and the bond coat is stronger than the bond coat itself [49]. Adhesive failure is dominant for the unmodified EABC at room temperature. With the inclusion of GO, the area of cohesive failure increases with the GO loading. In addition, cohesive failure becomes dominant when the GO loading is over 0.5 wt%. After pull-off tests at 60 °C, the unmodified EABC exhibits nearly all adhesive failures, as shown in Figure 13b. With the inclusion of GO, cohesive failure appears, and its area increases with the GO loading. Therefore, the incorporation of GO improves the pull-off strength of the unmodified EABC, and the pull-off strength of the modified EABCs increases with the GO loading. The improvement of the pull-off strength of the unmodified EABC is due to the interfacial reactions between GO and the substrate (steel plate) since GO contains many oxygen functional groups. With the increase in the GO loading, the quantity of functional groups increases. In this case, the interaction between GO and the substrate increases, and thus, the area of cohesive failure increases. Notably, the area of cohesive failure at 60 °C is much smaller than that at room temperature at the same GO loading. Therefore, the pull-off strengths of the unmodified and GO-modified EABCs at 60 °C are much lower than those at room temperature.

### 3.7. Shear Strength

Figure 14 shows the shear strengths of the unmodified and GO-modified EABCs. Like the pull-off strength, the presence of GO increases the shear strength at both room temperature and 60 °C. Furthermore, the shear strength of the modified EABCs increases with the GO loading. With the inclusion of 1.0 wt% GO, the shear strengths at room temperature and at 60 °C increase by 15% and 42%, respectively. Like the pull-off strength, the improvement of the shear strength of the unmodified EABC is also attributed to the interfacial reactions between the functional groups of GO and the substrate. Furthermore, the interaction between GO and the substrate increases with the GO loading. Consequently, the shear strength of the modified EABCs increases with the GO loading.

### 3.8. Mechanical Properties

The mechanical properties of the unmodified and GO-modified EABCs are shown in Figure 15. The presence of GO increases both the tensile strength and elongation at break of the unmodified EABC (Figure 15a). Additionally, the tensile strength and elongation at break of the modified EABCs increase with the GO loading. With the inclusion of 1.0 wt% GO, the tensile strength and elongation at break of the unmodified EABC increase by 21% and 38%, respectively.

Tensile toughness (area under the stress versus strain curve) represents the energy needed to fracture a material [50,51,52]. As illustrated in Figure 15b, the presence of GO increases the tensile toughness of the unmodified EABC. Among all modified EABCs, the one with 0.5 wt% GO has the maximum toughness (7.92 MJ/m^3^), which is 22% higher than that of the unmodified EABC (6.47 MJ/m^3^).

Overall, the inclusion of GO enhances the mechanical properties of the unmodified EABC. The improvement of the mechanical performance of the unmodified EABC is attributed to the uniform dispersion of GO in the epoxy asphalt and the good interfacial reactions between GO and epoxy asphalt. In addition, the quality of functional groups increases with the GO loading. Thus, like the bonding strength, both the tensile strength and elongation at break increase with the GO loading.

## 4. Conclusions

To improve the bonding strength of EABCs, graphene oxide was introduced. XRD analysis shows that GO nanolayers were exfoliated in the EABC. The inclusion of GO improved the hydrophobicity of the unmodified EABC. The inclusion of GO lowered the viscosity of the unmodified EABC during curing and thus extended the allowable construction time of the unmodified EABC. In addition, the allowable construction time of the modified EABCs increased with the GO loading. The presence of GO increased the storage modulus of the unmodified EABC in the glassy and rubbery states. The T_g_ of asphalt of the unmodified EABC was lowered with the addition of GO. The bonding strength of the unmodified EABC, especially at 60 °C, was enhanced with the incorporation of GO. In addition, the bonding strength of the modified EABCs increased with the GO loading. With the inclusion of GO, the area of cohesive failure of the unmodified EABC increased, while the area of adhesive failure decreased. For the modified EABCs, the area of cohesive failure increased with the GO loading, whereas the area of adhesive failure showed the opposite trend. The incorporation of GO improved the mechanical properties of the unmodified EABC. Further, the tensile strength and elongation at break of the modified EABCs increased with the GO loading.

## Figures and Tables

**Figure 1 materials-15-06846-f001:**
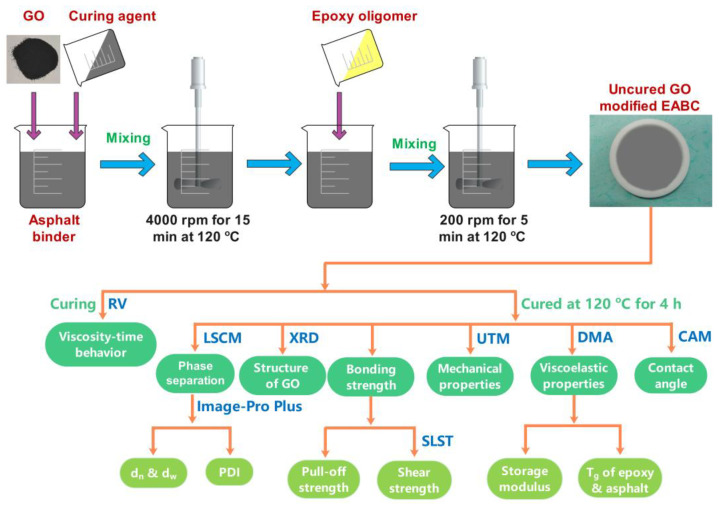
Schematic illustration for the preparation and characterization of GO-modified EABCs.

**Figure 2 materials-15-06846-f002:**
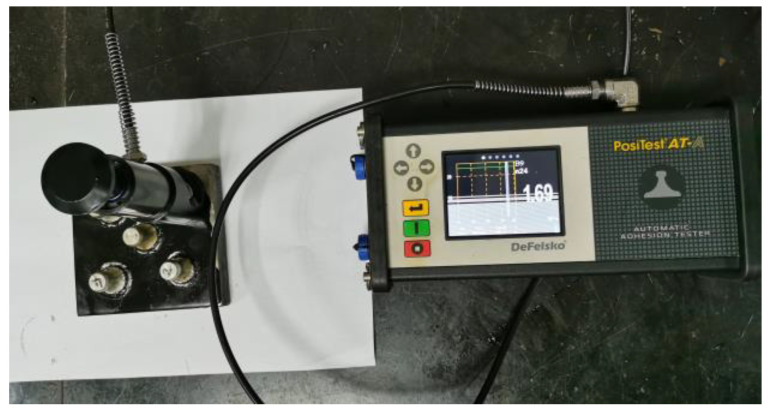
Portable automatic adhesion tester and testing configuration.

**Figure 3 materials-15-06846-f003:**
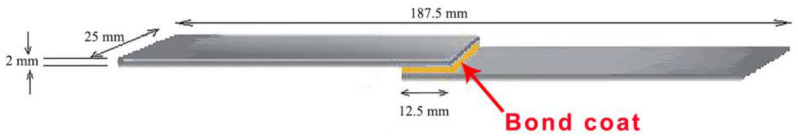
Schematic illustration of the sample for the single-lap shear test.

**Figure 4 materials-15-06846-f004:**
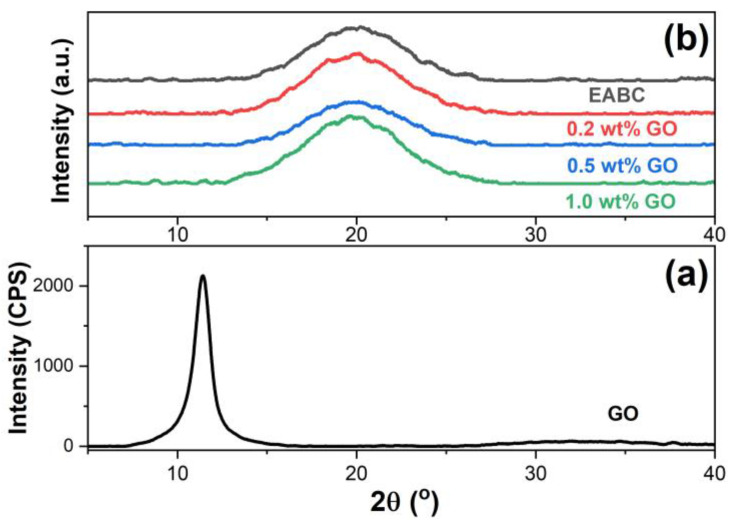
XRD patterns of pristine GO (**a**) and the unmodified and GO-modified EABCs (**b**).

**Figure 5 materials-15-06846-f005:**
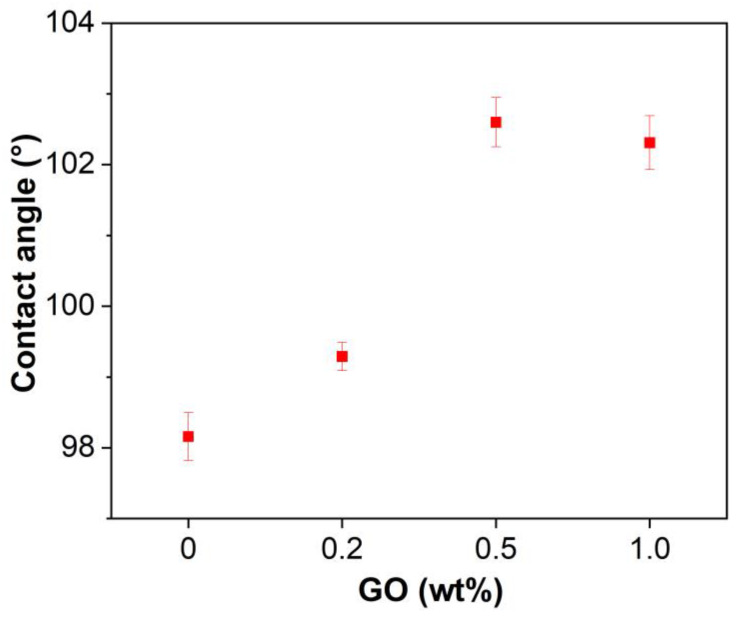
Contact angles of the unmodified and GO-modified EABCs.

**Figure 6 materials-15-06846-f006:**
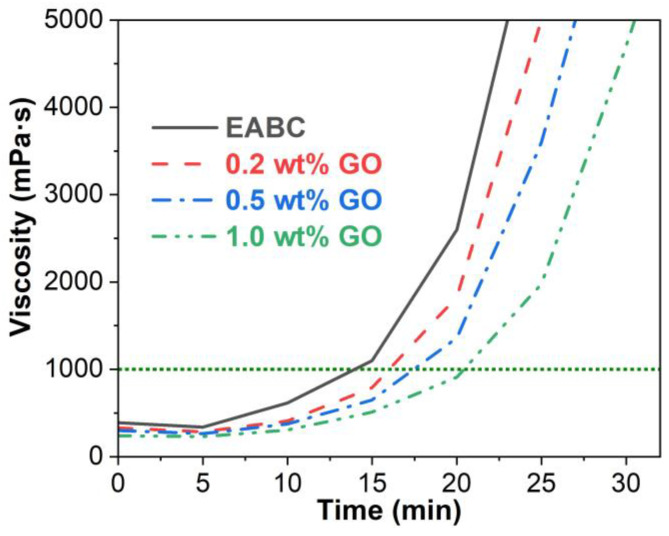
Viscosity–time curves of the unmodified and GO-modified EABCs at 120 °C.

**Figure 7 materials-15-06846-f007:**
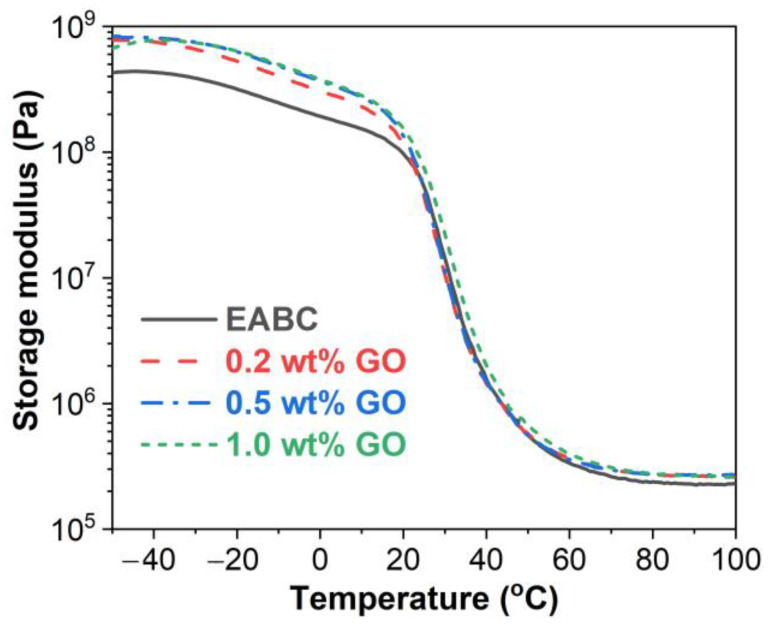
Storage modulus versus temperature curves of the unmodified and GO-modified EABCs.

**Figure 8 materials-15-06846-f008:**
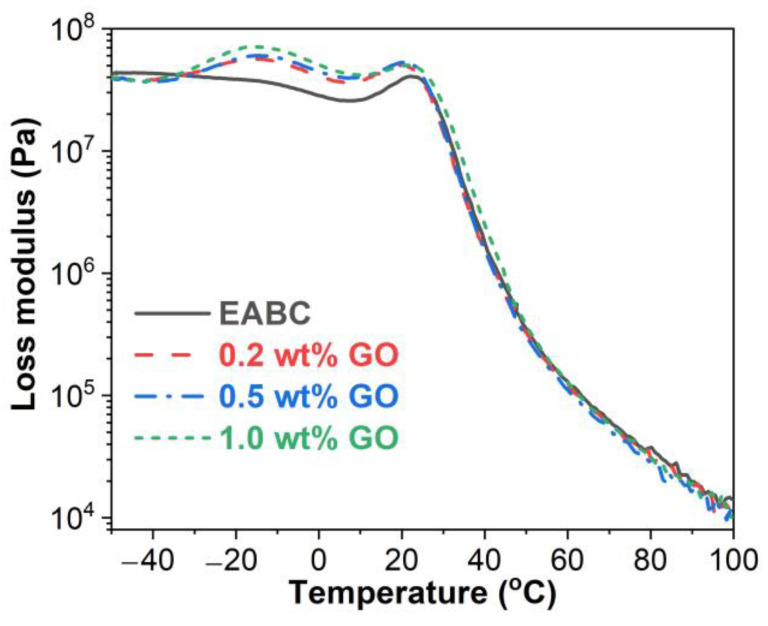
Loss modulus versus temperature curves of the unmodified and GO-modified EABCs.

**Figure 9 materials-15-06846-f009:**
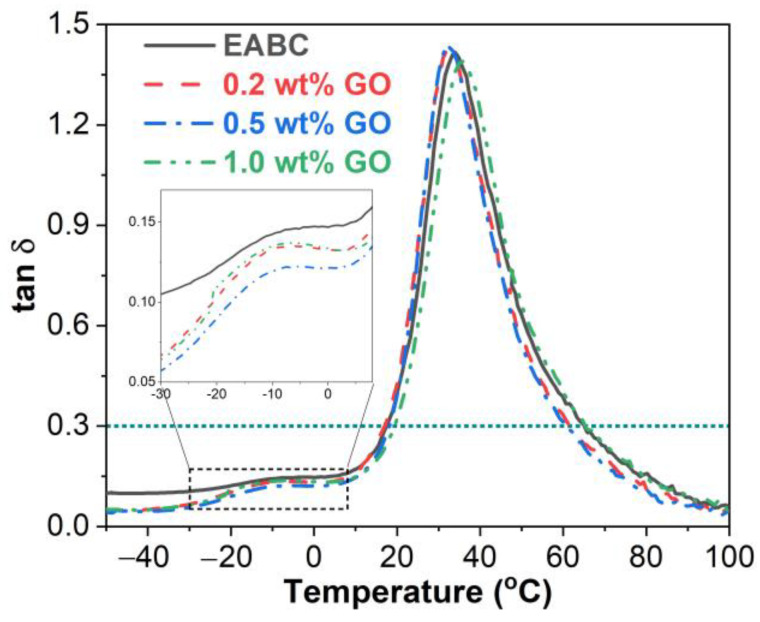
Loss factor versus temperature curves of the unmodified and GO-modified EABCs.

**Figure 10 materials-15-06846-f010:**
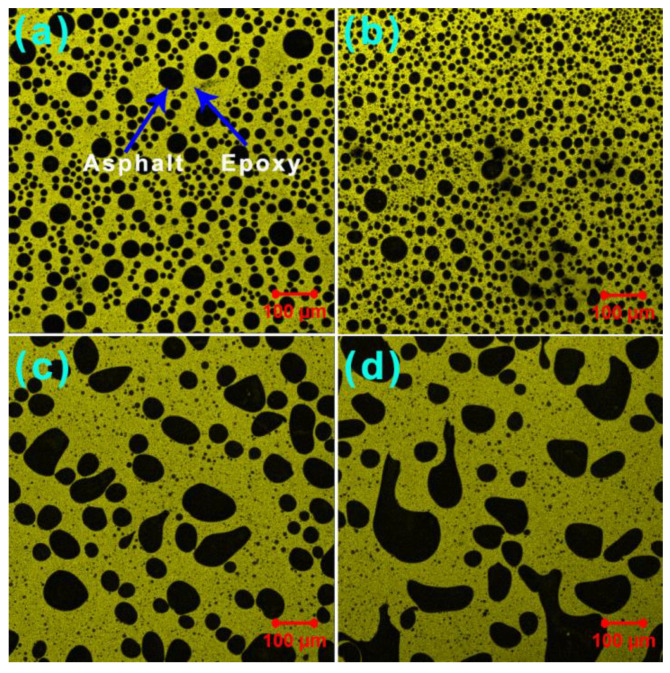
Fluorescence confocal microscopy photographs of the unmodified EABC (**a**) and GO-modified EABCs: 0.2 wt% (**b**), 0.5 wt% (**c**) and 1.0 wt% (**d**).

**Figure 11 materials-15-06846-f011:**
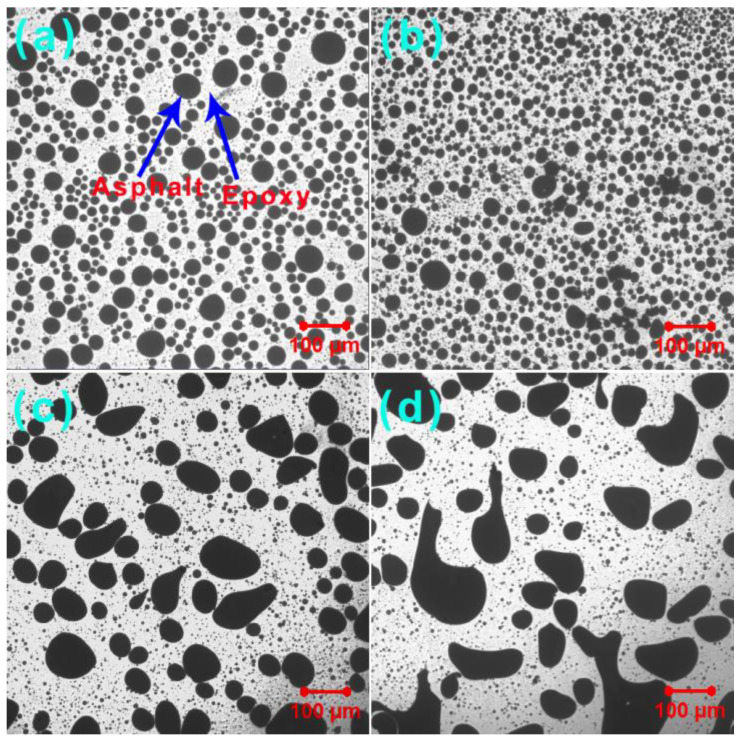
Transmittance confocal microscopy photographs of the unmodified EABC (**a**) and GO-modified EABCs: 0.2 wt% (**b**), 0.5 wt% (**c**) and 1.0 wt% (**d**).

**Figure 12 materials-15-06846-f012:**
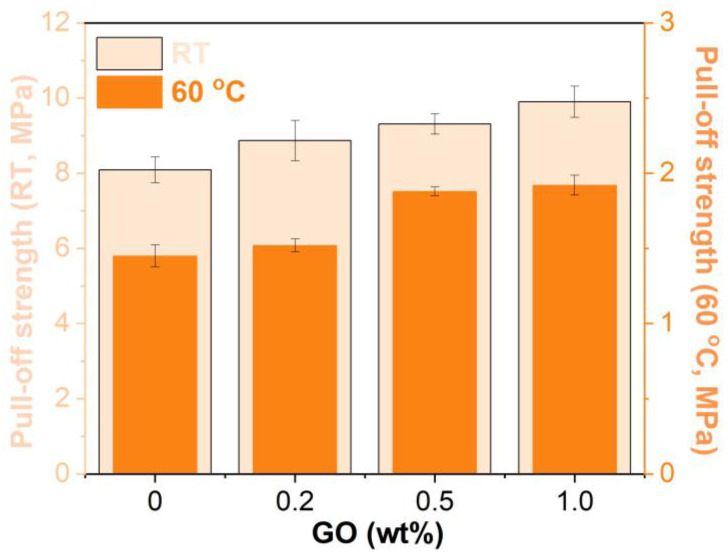
Pull-off strengths of the unmodified and GO-modified EABCs.

**Figure 13 materials-15-06846-f013:**
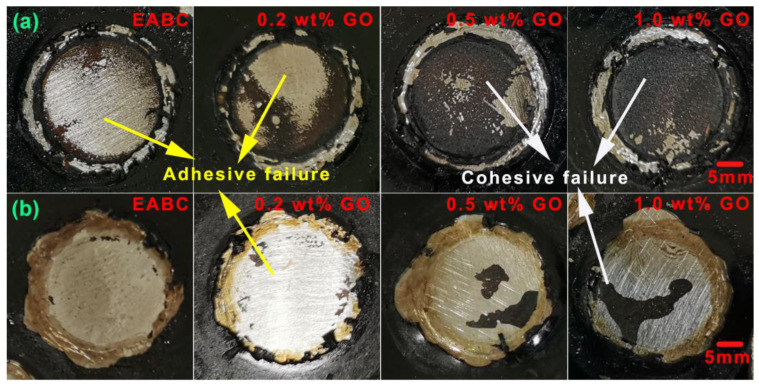
Failure surfaces of the unmodified and GO-modified EABCs on steel plates after pull-off tests at room temperature (**a**) and 60 °C (**b**).

**Figure 14 materials-15-06846-f014:**
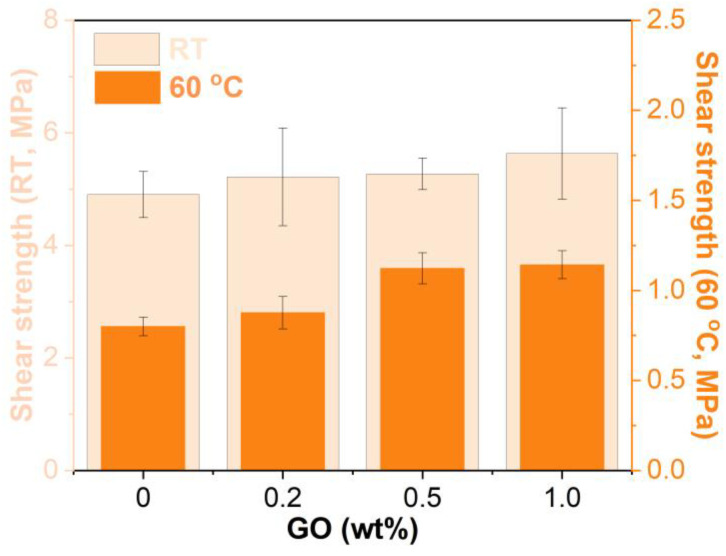
Shear strengths of the unmodified and GO-modified EABCs.

**Figure 15 materials-15-06846-f015:**
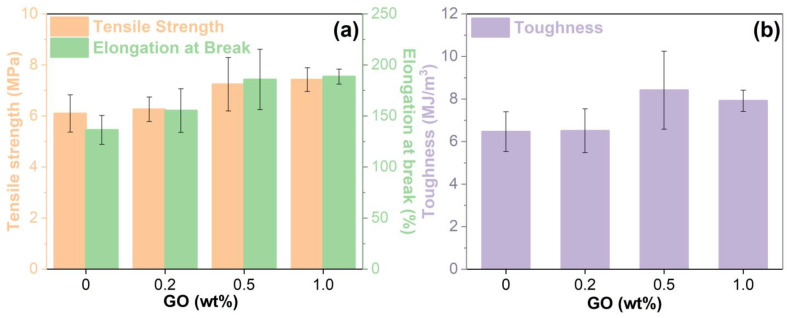
Mechanical properties of the unmodified and GO-modified EABCs: tensile strength, elongation at break (**a**) and toughness (**b**).

**Table 1 materials-15-06846-t001:** Overview of base asphalt.

Property	Standard	Value
Penetration (25 °C, 0.1 mm)	ASTM D5-06	91.0
Ductility (10 °C, cm)	ASTM D113-07	93.0
Softening point (°C)	ASTM D36-06	46.3
Viscosity (120 °C, mPa·s)	ASTM D4402-06	787
Saturates (%)	ASTM D4124-09	16.7
Aromatics (%)		33.9
Resins (%)		44.7
Asphaltenes (%)		4.7

**Table 2 materials-15-06846-t002:** Overview of graphene oxide powder.

Property	Value
Number of layers	1~2
Diameter (µm)	10~50
Thickness (nm)	~1
Carbon content (%)	<50
Oxygen content (%)	>42
Sulfur content (%)	<4
Purity (%)	95

**Table 3 materials-15-06846-t003:** T_g_s of asphalt and epoxy in the unmodified and GO-modified EABCs.

GO (wt%)	T_g_ of asphalt (°C)		T_g_ of epoxy (°C)	
	E”	tan δ	E”	tan δ
0	−13.7	−8.6	22.1	34.0
0.2	−14.4	−10.2	20.6	32.5
0.5	−14.4	−10.9	20.8	31.8
1.0	−16.2	−9.4	21.4	35.9

**Table 4 materials-15-06846-t004:** Average diameters and PDIs of dispersed asphalt domains in the unmodified and GO-modified EABCs.

GO (wt%)	*d_n_* (mm)	*d_w_* (mm)	*PDI*
0	23.1	34.0	1.47
0.2	15.6	18.9	1.21
0.5	17.3	51.0	2.95
1.0	17.3	56.6	3.27

## Data Availability

All data are available in the manuscript.

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
