# Peer review of "Graphene Oxide-Modified Epoxy Asphalt Bond Coats with Enhanced Bonding Properties"

_materials, 2022, doi:10.3390/ma15196846_

Round 1

Reviewer 1 Report

1. Abstract is good but still need to improve further. Please include  novelty at the end of abstract.  Please highlight main problem statement and also please add significant finding.

2. AVOID to use pronoun in article (Example: they, our etc.)

3. Idea to add GO in EABC its relevant or practical to the highway industry? Please justify this

4. Line 50 ‘EABC and epoxy…’- Please avoid to start sentences with Abbreviation/Number/Symbol (Check for whole article)

5. In method - please explain how GO distribution will control

6. Line 180 - 3.1. XRD(Please add details name of XRD first

7. Figure 4 lack of discussion. Please add more discussion and justification

8. Line 185-191: This paragraph look refer to much with previous reference. Please highlight the finding or idea from this study first. Strongly suggest to REVISE and reduce citation from previous - this show lack of novelty.

9.  Figure 5 lack of discussion. Please add more discussion and justification

10. Line 202-2091: This paragraph look refer to much with previous reference. Please highlight the finding or idea from this study first.Strongly suggest to REVISE and reduce citation from previous - this show lack of novelty.

11. Writing style of  results and discussion need to REVISE. Please avoid to cite any reference on early discussion. This show the article lack of novelty. PLEASE REVISE

12. Figure 10 & 11 - please add some labelling related to discussion

13. Line 314 - ‘The presence of GO increases the pull-off strength..’ Please justify this statement and how this will happen.

14. Line 335 - ‘With the inclusion of GO, the area of cohesive failure increases with the GO loading..’ This statement need more critical discussion and justification. How it happen?

15. Figure 14 and 15 lack of discussion. Please add more discussion and justification

16. Overall Discussion need to IMPROVE with more CRITICAL DISCUSSION

17. Conclusion too long and  need to mapping with problem statement in Introduction. Please REVISE

Reviewer 2 Report

The submitted manuscript highlights the evaluation of different properties of pure EABC as compared to 0.2wt%, 0.5wt% and 1wt% GO. The manuscript has been well written with all the measurement information adequately provided. The manuscript can be accepted for publication after addressing the minor points listed below

  • The resolution of all the figures (especially texts in all the figures) should be improved. They look highly pixelated when viewed in PDF version with some background (looks like copy paste). Schematics are fine though.

  • For the Storage modulus studies presented by the authors, it is clearly evident that the addition of 0.2 wt% and 0.5 wt% GO do not lead to any appreciable increase of the storage modulus of the EABC. How does GO performs infront of other additives. Did the authors tried any more than 1 wt% of GO concentration and see how it affects the storage modulus of EABC. The wt% of GO do not seem to be optimized for the storage modulus studies.

  • Similar observations go for the GO-EABC composite when it comes to loss modulus. Not very pronounced effect is observed as compared to pure EABC. Authors must compare this finding as compared to other additives which have been tried with EABC and therefore present an outline for the scope of GO in these regimes. The concentration optimization is still questionable.

  • If possible, One TEM image should also be provided to fully confirm the presence of GO flakes / sheets in the composite.

  • The other comments are mentioned in the attached pdf.

Reviewer 3 Report

Appreciation

The work is interesting, with novel results regarding the effect of graphene oxide (GO) on the wear resistance and hardness (adhesion degree) of the epoxy asphalt bonding layer (EABC).

Findings

1. The technology for the preparation of the tested material, the methodology and the apparatus for investigating the morphology and physical-mechanical properties are adequate

2. Experimental results, based on diagrams and micro-structure images are correctly interpreted

3. the use of the professional software Image-Pro Plus ensures a very good analysis with pertinent conclusions

Conclusion

The paper has scientific value and I recommend acceptance

Author Response

Thank you.

Round 2

Reviewer 1 Report

Author has made some improvement on this article, but some more improvement need to provide:

1. This statement '...provides the feasibility improving the bonding performance with graphene oxide.' need to add more discussion and justification in Results and discussion with simple schematic. 

2. Please avoid start sentences with Abbreviation or short form. Please check for whole article.

3. Figure 10, 11, 13 text labelling not clear. Please improve
